# *ZNF703* mRNA-Targeting Antisense Oligonucleotide Blocks Cell Proliferation and Induces Apoptosis in Breast Cancer Cell Lines

**DOI:** 10.3390/pharmaceutics15071930

**Published:** 2023-07-11

**Authors:** Sandra Udu-Ituma, José Adélaïde, Thi Khanh Le, Kenneth Omabe, Pascal Finetti, Clément Paris, Arnaud Guille, François Bertucci, Daniel Birnbaum, Palma Rocchi, Max Chaffanet

**Affiliations:** 1Equipe Labellisée Ligue Nationale Contre le Cancer, Predictive Oncology Laboratory, Marseille Research Cancer Center, INSERM U1068, CNRS U7258, Institut Paoli-Calmettes, Aix Marseille University, 13009 Marseille, France; sandra.udu-ituma@inserm.fr (S.U.-I.); adelaidej@ipc.unicancer.fr (J.A.); khanh.le-thi@inserm.fr (T.K.L.); keomabe@coh.org (K.O.); finettip@ipc.unicancer.fr (P.F.); clemmentparis@gmail.com (C.P.); guillea@ipc.unicancer.fr (A.G.); bertuccif@ipc.unicancer.fr (F.B.); palma.rocchi@inserm.fr (P.R.); 2Department of Biology, Alex Ekwueme Federal University Ndufu-Alike Ikwo, Abakaliki P.M.B. 1010, Ebonyi State, Nigeria; 3European Center for Research in Medical Imaging, Aix-Marseille University, 13005 Marseille, France

**Keywords:** *ZNF703*, luminal B breast cancer, antisense oligonucleotides, apoptosis, cisplatin

## Abstract

The luminal B molecular subtype of breast cancers (BC) accounts for more than a third of BCs and is associated with aggressive clinical behavior and poor prognosis. The use of endocrine therapy in BC treatment has significantly contributed to the decrease in the number of deaths in recent years. However, most BC patients with prolonged exposure to estrogen receptor (ER) selective modulators such as tamoxifen develop resistance and become non-responsive over time. Recent studies have implicated overexpression of the *ZNF703* gene in BC resistance to endocrine drugs, thereby highlighting *ZNF703* inhibition as an attractive modality in BC treatment, especially luminal B BCs. However, there is no known inhibitor of *ZNF703* due to its nuclear association and non-enzymatic activity. Here, we have developed an antisense oligonucleotide (ASO) against *ZNF703* mRNA and shown that it downregulates *ZNF703* protein expression. *ZNF703* inhibition decreased cell proliferation and induced apoptosis. Combined with cisplatin, the anti-cancer effects of *ZNF703*-ASO9 were improved. Moreover, our work shows that ASO technology may be used to increase the number of targetable cancer genes.

## 1. Introduction

Breast cancer (BC) is the most common type of cancer and the main cause of cancer death in women [1]. Despite a better understanding of biology and improvements in prognosis and treatment, BC remains associated with morbidity and mortality. There is a need for more effective systemic treatments. Based on the estrogen receptor (ER) and ERBB2/HER2 receptor four molecularly distinct BC subtypes can be distinguished [2] i.e., luminal A/ER+, luminal B/ER+, ERBB2/HER2+, and basal/triple-negative [3]. Over the years, considerable efforts have focused on identifying reliable prognostic markers and therapeutic targets for each of these subtypes.

Therapeutic targeting of ER has significantly contributed to the decrease in the number of deaths in recent years. Tamoxifen is a selective ER modulator (SERM). Its use as adjuvant systemic therapy in hormone receptor-positive BC patients increases the 15-year survival rate. [4]. However, resistance to endocrine therapy remains a significant cause of death for these patients [5].

BC patients with prolonged exposure to endocrine therapy develop resistance and become non-responsive. Progress has been made in understanding the underlying resistance mechanisms, but most remain unknown. Auto-activation of the ER (*ESR1*) gene due to gain-of-function mutation is a major cause of resistance to aromatase inhibitors (AI) [6,7]. However, other mechanisms of resistance have been identified [8] in advanced/metastatic BCs. In addition, intrinsic resistance to hormonal therapies can be observed at diagnosis [9]. Here, failure of endocrine therapy might be explained by the activation of signaling pathways (PI3K/AKT/MTOR, ERBB2/HER2, FGFR, CDK4/CDK6, MDM2–TP53 interaction, Histone deacetylases) that crosstalk with the ER pathway [8]. Recent studies have also shown the putative involvement of various proteins (e.g., CITED2, NCOR2, AGR2, FGFR4, FOXA1) and molecular networks [10,11,12,13,14,15,16,17,18].

Numerous studies of endocrine resistance in ER-positive tumors have been reported [8,19,20]. A poor prognosis is associated with these tumors which are aggressive and often do not respond well to neoadjuvant chemotherapy [20,21,22,23]. Several therapy strategies have been established (i) optimization of endocrine therapy administration (e.g., fulvestrant), (ii) combination of different endocrine therapies (anastrozole, letrozole, fulvestrant), (iii) of endocrine therapy with anti-tyrosine kinases (e.g., anti-FGFR, -IGF1R …), (iv) of endocrine therapy with CDK inhibitors (palbociclib…), (v) of endocrine therapy with agents targeting the PI3K/AKT/MTOR pathway (alpelisib, everolimus) and steroid sulfatase inhibitors. Cyclin-dependent kinase 4 and 6 (CDK4/6) inhibitors (e.g., palbociclib, ribociclib) have shown significant efficacy in ER+ BC [24,25]. However, their effects are also limited by drug resistance [26,27].

Luminal B BCs account for up to 39% of BCs [28]. They are associated with aggressive clinical behavior and poor prognosis. Although they express hormone receptors, they have a high risk of metastasis and resistance to both hormone and conventional chemical therapies [29]. The genomic landscape of luminal B BCs shows significant chromosomal alterations associated with 8p11-p12 and 11q13.1-q13.2 amplifications, and specific gains and losses [30] as well as various mutated genes, *TP53* and *PIK3CA* being the most frequent [31]. *BRCA2*-mutated tumors display the luminal B subtype [32].

Our laboratory has contributed to a better understanding of luminal B BCs and reported several luminal B candidate genes that may play a role in this aggressive subtype’s development and/or hormone resistance [30,33,34,35]. The *ZNF703* oncogene may represent a potential actionable target among the luminal B candidate genes. *ZNF703* is located in the 8p12 chromosomal region, which is specifically amplified in luminal B BC [30,34,36,37,38]. Increased BC risk was recently correlated with upregulation of *ZNF703* [39]. The biology of the *ZNF703* protein is not well known yet. *ZNF703* is a transcriptional corepressor that regulates many genes involved in multiple aspects of the cancer phenotype, such as proliferation, invasion, and altered balance of stem cells. *ZNF703* overexpression influences several aspects of the pathology of BC, including ER signaling and the phenotypes of progenitor cells [34,38,40]. High expression of *ZNF703* is associated with poor prognosis in BC patients [34,38,41]. Beside luminal B BC, high expression of *ZNF703* was also found by immunohistochemistry in a third of TNBC patients [42] as well as in other cancers such as gastric, colorectal, head and neck cancers, non-small cell lung, cholangiocarcinoma, papillary thyroid carcinoma (PTC), medullary thyroid carcinoma (MTC), hepatocellular carcinoma (HCC) and ovarian cancer [43,44,45,46,47,48,49,50,51]. *ZNF703* might play a role in tumor metastasis by promoting epithelial-mesenchymal transition (EMT) through the repression of E-cadherin expression [40]. A recent study reported that *ZNF703* overexpression induces EMT and sorafenib resistance in HCC by transactivating CLDN4 expression [51]. These results suggest that pathways regulated by *ZNF703* could be targeted by personalized therapy alone or in combination with other drugs.

However, *ZNF703* is not an enzyme and functions as a transcriptional corepressor and there is neither direct nor indirect *ZNF703* inhibitor.

Antisense oligonucleotides (ASOs) are short nucleic acid sequences, usually single-stranded deoxyribonucleotides (DNA) (length ~ 20 bp) that are complementary to the target mRNA. Hybridization of the ASO to the target mRNA via Watson-Crick base pairing can result in the specific inhibition of gene expression through various mechanisms, depending on the chemical composition of the ASO and location of hybridization, thereby blocking the translation of the mRNA [52]. ASO use has helped study the loss of gene function and target validation. It has become highly valuable as a novel therapeutic strategy for diseases linked to dysregulated gene expression such as cancer and regenerative diseases [53]. We have previously developed and patented several ASOs targeting different genes (*HSP27*, *TCTP*, *MEN1*, *DDX5*) implicated in different cancers, including prostate cancer [54,55,56,57,58,59,60]. Therefore, targeting *ZNF703* by ASO could represent a promising approach in the treatment of luminal B BCs.

In this work, we have developed an ASO targeting the mRNA of *ZNF703* and evaluated the therapeutic efficacy of ASO-based *ZNF703* inhibition in BC cell lines.

## 2. Materials and Methods

Cell Lines. Fifteen established breast tumor cell lines (ATCC, Rockville, MD, USA; [61]), subtyped by PAM50 [62] as luminal A (LA) (BT483, HCC1500, MDA-MB-134), luminal B (LB) (MCF7, SUM185, T47D), ERBB2 (ERBB2) (HCC1954) basal (B) (HCC38, HCC1569, HCC1806, MDA-MB-231, SKBR-7, SUM149, 184B5), normal-like (N-L) (MCF-10F), (Thermo Fisher Scientific; Life Technologies SAS, Courtaboeuf, France) as well as differently engineered luminal B MCF7 BC cell lines (MCF7-GFP and MCF7-*ZNF703*/GFP, respectively) [34] were studied for *ZNF703* mRNA and protein expressions. Cells were grown using the recommended culture conditions. The MCF-7-GFP-and MCF-7-*ZNF703*/GFP engineered cell lines were maintained in RPMI 1640 medium (Life Technologies SAS, Courtaboeuf, France), supplemented with 10% fetal bovine serum (FBS), 0.5% human insulin (2 mg/mL), and 1% Hepes 1 M, 1% Antibiotic-Antimycotic (100X), 1% amino acid (Life Technologies SAS, Courtaboeuf, France). These cell lines were maintained at 37 °C in a 5% CO_2_ humidified atmosphere.

Design and Synthesis of Antisense Oligonucleotides. To design antisense oligonucleotides (ASOs) targeting the entire *ZNF703* mRNA, an R-based software was developed in our laboratory (PDA16130, 2017) as previously described [58]. First, the coding portion of the target transcript was selected and segmented into consecutive sequences of 20 bases. Subsequently, to define potential ASOs, the complementary sequences of the resulting sequences were identified and reversed to the 5′-3′ direction. The program’s output gave information about the ASO list sequences with their GC content and genes list with significant similarity. The final selection of *ZNF703*-ASOs was made manually, excluding ASOs showing similarity to other genes. Fourteen *ZNF703*-ASOs (Appendix A) were designed and synthesized on OligoPilot 10 automated DNA synthesizer (50 μmol scale). The length of all 14 synthesized *ZNF703*-ASOs was 20 bp. The oligonucleotide synthesis was done using standard β-cyanoethyl phosphoramidite chemistry. Oligonucleotide sequences were fully modified with a phosphorothiated (PS) backbone to protect them from nuclease degradation. After synthesis, oligonucleotide cleavage and deportation were done in concentrated ammonium hydroxide at 55 °C for 16 h. Purification was done on ion pairing reversed-phase high-pressure liquid chromatography (IP-RP HPLC). The purity was assessed by analytical IP-RP-HPLC and characterized by MaldiTof mass spectrometry. The *ZNF703*-ASO sequence corresponding to the human *ZNF703* mRNA at position 390-409 was 5′-GGTGTGAGCGCTCAGCATCT-3′. The scrambled (SCR) control sequence was 5′ CGTGTAGGTACGGCAGATC-3′ and designated as control/scrambled-ASO.

Transfection with Antisense Oligofectamine (ASOs). Cells were plated at a density of 60–80% and transfected twice (24 h and 48 h) after seeding with ASOs. To enhance cell uptake and cellular trafficking of ASOs during ASOs treatment, oligofectamine (a cationic lipid-transfection reagent) was used (Invitrogen, Life Technologies, Burlington, ON, Canada). Cells were treated with *ZNF703*- or scrambled-ASOs (50 to 300 nM) after pre-incubation for 20 min with 3mg/mL oligofectamine in serum-free OPTI-MEM (Life Technologies, AS, Courtaboeuf, France). Following 4h incubation, the medium containing ASOs and oligofectamine was replaced with the complete medium. On the following day, the same ASO treatment was done. ASO transfections were done twice to increase its internalization and its efficiency in the cells.

*ZNF703* mRNA expression. We had previously established the mRNA profiles of 31 BC cell lines using whole-genome DNA microarrays (HG-U133 Plus 2.0, Affymetrix) and Robust Multichip Average (RMA) method in R using Bioconductor and associated packages [63,64]. We interrogated *ZNF703* expression data and centered it on those observed in the HME-1 cell line. Fifteen BC cell lines were chosen for the present study regarding their *ZNF703* mRNA expression defined as follows: (i) top 6 of upregulated; (ii) 3 medium and (iii) top 6 of downregulated (see Figure 1A).

Western Blot. *ZNF703* protein expression was assessed by using WB analysis as previously described [58]. 72h before WB analysis, BC cells (transfected or not with ASOs or Scramble) were lysed and protein content was prepared and quantified. For each sample, 40 μg of protein were used to prepare WB. For immunodetection, the membranes were incubated at 4 °C with 1:3000 rabbit anti-*ZNF703* polyclonal antibody (GTX107721 GeneTex Inc., Euromedex 67460 Souffelweyersheim, France) and 1:1500 rabbit anti-GAPDH polyclonal antibody (ab9485 Abcam, Cambridge, UK) as endogenous loading controls. After incubation with horseradish-conjugated anti-rabbit secondary antibody (1:4000, 1 h, at room temperature), detection of the protein bands was accomplished using ECL Prime Western Blotting detection reagent (RPN2236, GE Healthcare, Vélizy-Villacoublay, France) and developed on Amersham Hyper film ECL films (GE Healthcare Buckinghamshire, UK).

Cell Viability with MTT Assay. MCF7-*ZNF703*/GFP-cells were plated in 12-well plates (3 × 10^5^ cells/well) maintained in culture media (RPMI) for 48 h and transfected the day after with *ZNF703*- or scrambled-ASO at 50 nM, 70 nM, 100 nM, 150 nM, and 200 nM. After 72 h, subjected to a cell viability test, MTT (3-(4,5-dimethylthiazol-2yl)-2,5-diphenyl tetrazolium) was added to each well (1 mg/mL; final concentration) and the plates were incubated for 2–3 h at 37 °C. Supernatants were then removed and formazan crystals, formed as a result of the enzymatic reduction of MTT, were dissolved in 500 μL of Dimethyl sulfoxide (DMSO). The absorbance (595 nM) was evaluated using a Sunrise microplate absorbance reader (Tecan, 69003 Lyon, France). Each assay was done in triplicate. Cell viability was expressed as the percentage of absorbance of transfected cells compared to untreated cells.

Cell Treatment with cisplatin and MTT Assay. MCF7-*ZNF703*/GFP cells were seeded in 12-well plates with 30,000 cells/well and transfected on the day after seeding with 100 nM of ASO or Scrambled. The transfection was repeated on the next day. After 48 h, cells were then treated with 100 nM (half-maximal inhibitory concentration (IC50)) of cisplatin (Sanofi-Aventis, France) for 24 h. MTT treatment was done as described in previous chapter (Cell Viability with MTT Assay). Each assay was done in triplicate.

Cell Cycle Distribution Assay. MCF7-*ZNF703*/GFP cells (2 × 10^5^) were seeded in 100 mm culture dishes. On the following day, the cells were treated with *ZNF703*- or scrambled-ASO at 200 nM for 48–72 h. Cells preparation, DNA content examination by flow cytometry and calculation of percentage of cells in the G0, G1, S, and G2/M phases were done as previously described [58]. The assay was done in triplicate.

Cell Apoptosis by Annexin V Assay. MCF7-*ZNF703*/GFP cells were plated at the density of 10^5^ cells into 100 mm culture dishes. On the following day, cells were treated with 200 nM of *ZNF703*- or scrambled-ASO twice. After 72 h of incubation, cells were prepared and stained for cell apoptosis evaluation by APC Annexin V/Dead Cell Apoptosis Kit with APC annexin V and SYTOX Green for Flow Cytometry (Life Technologies SAS, Courtaboeuf, France). as previously described [58]. Rates of cell deaths were then measured using FlowJo (Becton Dickinson France SAS, Grenoble, France). The experiments were done in triplicates.

Statistical Analysis. Gel band density and migration distances were measured with ImageJ software (NIH). Statistical analysis was done using the Graph Pad Prism program (Graph Pad Software, San Diego, CA, USA). All data are mean values. All the results were expressed as mean ± SD. The significance of differences was assessed by a two-tailed Student’s *t*-test. * *p* ≤ 0.05 was considered significant, with ** *p* ≤ 0.01, *** *p* ≤ 0.001, and **** *p* < 0.0001.

## 3. Results

### 3.1. ZNF703 Expression in Different Breast Cancer Cell Lines

From the 31 BC cell lines with mRNA profiles, we previously reported [63], 15 were chosen in function of their *ZNF703* mRNA expression as follows: (i) top 6 of upregulated (High); (ii) 3 medium (Med) and (iii) top 6 of downregulated (Low) (Figure 1A). The *ZNF703* protein expression in these cell lines was assessed by western blotting (WB) and compared with their *ZNF703* mRNA expression (High, Med, and Low) (Figure 1B). Most of the cell lines had *ZNF703* protein expression in agreement with the cognate mRNA expression. High *ZNF703* expression was observed in BT483, HCC1500, MCF7, SKBR7, and T47D. *ZNF703* expression assessed by WB on flow cytometry sorted MCF7-*ZNF703*/GFP and MCF7-GFP cell lysates were used as controls (Figure 1C). We next collected and analyzed the gene expression and protein expression data of 316 cancer cell lines, including 28 breast cell lines, of the Broad Institute Cancer Cell Line Encyclopedia (CCLE) [67] hosted by the Cancer Dependency Portal (DepMap, 22Q4 version). A significant positive correlation between the mRNA and protein expression levels of *ZNF703* (Figure 1D) was established in all cell lines (Pearson r = 0.75, *p* = 3.66 × 10^−57^) and BC cell lines (Pearson r = 0.77, *p* = 1.43 × 10^−6^). This result indirectly suggests that the changes in gene expression in tumors are likely to translate into changes at the protein level for *ZNF703*.

### 3.2. ASO Design, Synthesis, and Screening for Inhibitory Activity on ZNF703 mRNA

Because *ZNF703* targeting may help fight *ZNF703*-overexpressing luminal B BCs, we developed an ASO against *ZNF703* mRNA to downregulate its protein expression level in BC cell lines.

We screened, by gene walk, all ASO sequences targeting *ZNF703* full-length mRNA. An initial set of 14 *ZNF703*-ASOs (ASO2, ASO9, ASO13, ASO17, ASO24, ASO26, ASO37, ASO51, ASO67, ASO68, ASO81, ASO83, ASO86 and ASO87) against *ZNF703* mRNA was then designed (Appendix A). The *ZNF703*-ASOs were evaluated by WB for their capacity to repress both endogenous and exogenous *ZNF703* (*ZNF703* and *ZNF703*/GFP) protein expression in MCF7-*ZNF703*/GFP BC cells at 100 nM concentration (Appendix A and Figure 2). MCF7-*ZNF703*/GFP was chosen because it is a luminal breast cancer cell line in which *ZNF703* was artificially overexpressed leading to various consequences including cell proliferation and resistance to tamoxifen previously reported and known to reflect luminal B phenotype [34]. As shown in Appendix A and Figure 2A, *ZNF703*-ASO9 was the most efficient ASO for silencing *ZNF703* and achieved more than 70% inhibition of endogenous *ZNF703* protein expression. *ZNF703*-ASO9 was used for further analyses.

### 3.3. ZNF703-ASO9 Targeting ZNF703 mRNA Downregulates ZNF703 Protein Expression in BC Cell Lines

We assessed the inhibitory activity of *ZNF703*-ASO9 on *ZNF703* expression in two other BC cell lines (MDA-MB-134, luminal, and MDA-MB-231, basal) to establish its consistency in effectively inhibiting its target gene (Figure 3). MDA-MB-134 is a luminal B breast cancer cell line with the highest *ZNF703* mRNA expression (Figure 1A). MDA-MB-231 is a triple negative breast cancer cell line and could be used for comparison.

Treatment with ASO9 (100 nM) caused a significant (>70%) inhibition of *ZNF703*/GFP but only 40% inhibition of the endo-*ZNF703* in MCF7-*ZNF703*/GFP probably due to the competition with exogenous *ZNF703*/GFP mRNA (showed as a control in Figure 3A). Strong inhibition of *ZNF703* (>70%) was also observed in MDA-MB-134 (Figure 3B), while the *ZNF703*-ASO9-treated MDA-MB-231 cell line exhibited lower *ZNF703* expression inhibition but still greater than 50% (Figure 3C).

### 3.4. ZNF703-ASO9 Has Antiproliferative Effect in BC Cell Lines

To determine the effect of *ZNF703*-ASO9 on both endogenous and exogenous *ZNF703* mRNA, MCF7-*ZNF703*/GFP cells were treated with increasing concentrations of *ZNF703*-ASO9 (Figure 4A,C). A dose-dependent inhibition of endogenous *ZNF703* expression in MCF7-*ZNF703*/GFP was observed with the highest inhibition efficiency identified at 300 nM. *ZNF703*/GFP (83 kDa band) expression was inhibited at 100 nM of *ZNF703*-ASO9 (Figure 4A,B). The expression of the endogenous *ZNF703* band at 58 kDa (expressed in all tested BC cell lines as well as in the engineered MCF7 cell lines) was inhibited by *ZNF703*-ASO9 following a dose-dependent manner (Figure 4A,C).

In the same experiment, we observed, in the presence of increasing concentrations of *ZNF703*-ASO9, a dose-dependent decrease in MCF7-*ZNF703*/GFP cell viability was measured with the MTT test (Figure 4D). After 72 h of incubation post-second treatment, increasing concentrations of *ZNF703*-ASO9 (50–200 nM) led to a gradual decrease in the fraction of viable cells. Cells either treated with control-ASO (SCR) or not treated showed a higher fraction of viable cells than *ZNF703*-ASO9-treated cells.

The antiproliferative activity of *ZNF703*-ASO9 was also evaluated in MCF7-GFP, MDA-MB-134, MDA-MB-231, and HCC1954 cell lines, and the corresponding IC50 was calculated (Appendix A). HCC1954 cell line was used as a negative control due to its very low *ZNF703* expression (see Figure 1). While MCF7-*ZNF703*/GFP exhibited the most sensitive response to *ZNF703*-ASO9 (112.7 nM), IC50 values of 122.6, 116.1, and 123 nM were established for MCF7-GFP, MDA-MB-134, and MDA-MB-231 (Appendix A), respectively.

Cell viability assessment of MCF7-*ZNF703*/GFP cells after treatment with 11 various *ZNF703*-ASOs (200 nM) against *ZNF703* mRNA showed that all exhibited a significant decrease in cell viability (Student’s *t*-test, *** *p* ≤ 0.001). *ZNF703*-ASO9 was the most efficient ASO (Appendix A). SCR treatment exhibited only weak toxicity.

Several reasons could explain various effects on cell viability and *ZNF703* protein level of different designed ASOs [68]. Among them: (i) the primary sequence, chemical nature, and structure of the ASO can have impacts on the interaction of PS-ASOs with specific proteins; (ii) the activity of PS-ASOs is strongly influenced by the association with both inter- and intracellular proteins; (iii) PS ASO protein interactions can affect many aspects of their performance, including distribution and tissue delivery, cellular uptake, intracellular trafficking, potency and toxicity.

These results show that *ZNF703* inhibition by *ZNF703*-ASO9 has significant antiproliferative activity in BC cell lines (>80% of cell proliferation inhibition).

### 3.5. ASO Inhibition of ZNF703 Induces Cell Death

We next wanted to determine if *ZNF703* inhibition affected cell viability by inducing cell death.

We first studied the impact of *ZNF703* inhibition on cell cycle progression using flow cytometry in MCF7-*ZNF703*/GFP, MDA-MB-134, and MDA-MB-231 cells, after treatment with or without *ZNF703*-ASO9 at 200 nM. A significant increase in cell death was observed in MCF7-*ZNF703*/GFP and MDA-MB-231 cells treated with *ZNF703*-ASO9 compared to cells treated with control-ASO.

We next analyzed whether the antiproliferative activity and cell death induction by *ZNF703*-ASO9 was due to apoptosis by measuring annexin V binding in MCF7-*ZNF703*/GFP and MDA-MB-231 cells. Flow cytometry was used to quantify the apoptotic rates (Figure 5). Flow cytometry analysis of annexin V expression showed that treatment with *ZNF703*-ASO9 significantly increased apoptosis of MCF7-*ZNF703*/GFP cells (*** *p* ≤ 0.001) and MDA-MB-231 cells (* *p* ≤ 0.05) compared to control-ASO (Figure 5A,B, respectively). MCF7-*ZNF703*/GFP and MDA-MB-231 cells not treated (NT) and treated with control-ASO showed a non-significant difference in the percentage of annexin V positive cells, suggesting low toxicity of control-ASO.

These results suggest that *ZNF703* plays a role in the maintenance of cell survival by apoptosis blockade and suggest that *ZNF703* inhibition using ASO could trigger apoptosis.

### 3.6. The Anti-Cancer Effects of ASO9 Are Improved When Combined with Cisplatin in MCF7-ZNF703/GFP BC Cell Line

We next evaluated the efficacy of cisplatin either alone or in combination with ASO9 treatment in MCF7-*ZNF703*/GFP BC cells. In other terms, because *ZNF703* is an oncogene originally identified in luminal BC [34,38], we wanted to see if inhibition of *ZNF703* expression combined with cisplatin could improve the therapeutic efficacy of *ZNF703*-ASO9 in our luminal B model (MCF7-*ZNF703*/GFP) [34].

We inhibited *ZNF703* mRNA expression with ASO9 (100 nM) and treated MCF7-*ZNF703*/GFP cell line with cisplatin (100 nM) 48h post *ZNF703*-ASO9 transfection. Cell viability was evaluated by the MTT test (Figure 6). Cisplatin alone or in combination with SCR reduced cell viability more efficiently than the absence of treatment (NT) or treatment with SCR alone (Student’s *t*-test, *** *p* ≤ 0.001) suggesting that MCF7-*ZNF703*/GFP cell line is cisplatin sensitive. It also showed that the anti-cancer ability of *ZNF703*-ASO9 was improved when combined with cisplatin (Student’s *t*-test, *** *p* ≤ 0.001) (Figure 6).

Thus, *ZNF703*-ASO9 could be used as a targeted therapy in combination with cisplatin to improve therapeutic approaches of luminal advanced breast cancer patients.

## 4. Discussion

In this study, we report the first steps in the development of an effective gene-based strategy (ASO) targeting the *ZNF703* mRNA in BC. From an initial set of 14 ASOs, we identified ASO9 as the most efficient for silencing *ZNF703* expression in an engineered breast cancer cell line. We show that *ZNF703* inhibition decreases cell proliferation and induces apoptosis in BC cell lines. We also report that the combination with cisplatin improved ASO9′s anti-cancer effects in MCF 7-*ZNF703*/GFP luminal BC cell line.

### 4.1. ZNF703 Oncogene Is a Potential Therapeutic Target in Breast and Other Advanced Cancers

*ZNF703* is a transcriptional corepressor that regulates many genes involved in multiple facets of the cancer phenotype, including proliferation, invasion, and regulation of stem cells. Commonly found amplified in luminal B breast cancer, *ZNF703* gene overexpression influences several aspects of the pathology of BC, including ER signaling and the phenotypes of progenitor cells [34,38,40]. *ZNF703* overexpression is associated with poor prognosis in luminal BC, colorectal cancers, head and neck squamous carcinomas, cholangiocarcinomas, and ovarian cancers [34,38,41,42,43,44,45,47,50]. *ZNF703* overexpression was also reported in other cancers such as gastric cancers, non-small cell lung, PTC, MTC, and HCC [44,46,47,48,49,51].

*ZNF703* overexpression increases cell proliferation, stimulates tumor migration/invasion, and is involved in endocrine and chemoresistance in luminal B BCs. *ZNF703* is a target gene of the ER transcription and suppresses ER expression in a negative feedback loop [69]. It also suppresses cell cycle inhibitors p27 and p15, which leads to upregulation of E2F1 and increased cell proliferation [34,38]. An indirect effect on proliferation is observed in cells expressing *ZNF703* by interference with the inhibitory functions of TGFβ signaling [38]. *ZNF703* might also play a role in tumor metastasis by promoting EMT through the repression of E-cadherin expression [40].

Thus, *ZNF703* overexpression facilitates tumorigenesis, metastatic invasion, and predicts poor prognosis in various advanced cancers. It may be a potential therapeutic target for various cancers.

### 4.2. ASO Targeting ZNF703 mRNA May Be an Effective Strategy for Advanced BCs

There is no known direct or indirect inhibitor of *ZNF703*. *ZNF703* is neither an enzyme nor a surface receptor and thus not easily targetable. Moreover, as a nuclear protein, it has poor bio-accessibility. We thus developed an ASO strategy to downregulate *ZNF703* expression.

The use of nucleic acid-based technologies for gene silencing is increasingly taking center stage in therapy. Recently, over ten nucleic acid-based drugs have received FDA approval for the treatment of various diseases with several others currently at various stages of clinical trials [70,71]. ASOs have a long history of clinical development with eight approved ASOs since 1998 [72,73].

Several works have validated the therapeutic efficacy of ASOs in different cancer types including prostate cancer [54,55,56,57,59,60], ovarian cancer [74], and breast cancer [58].

We designed and developed an ASO against different *ZNF703* sequences and screened their efficacies to inhibit *ZNF703* at the mRNA and protein levels. ASO9 reduced *ZNF703* by 70% at 100 nM. An increase in the dose of ASO resulted in an increased inhibitory effect on *ZNF703* expression. *ZNF703*-ASO9 did not only inhibit the intrinsic expression of its target but also the engineered expression of *ZNF703*. *ZNF703* protein and mRNA expression correlated well, which was good news for ASO use.

*ZNF703* has been identified as a driver of the 8p12 amplification in luminal B BC. High expression of *ZNF703* luminal B tumor patients is associated with poor clinical outcome [34,41].

We found that a dose-dependent inhibition of exogenous *ZNF703* in our MCF7 model overexpressing *ZNF703* was directly correlated with a dose-dependent decrease in cell viability. Proliferation inhibition may be explained by cell cycle blockade and cell death induction via an apoptotic pathway. Such results are in agreement with previous studies using small interfering RNA (*ZNF703*-siRNA) [42,43,48,49]. *ZNF703* inhibition suppressed cell proliferation and blocked the cell cycle in BT-549 and MDA-MB-468 basal BC (TNBC) cells [42]. A G1-phase arrest was induced by *ZNF703* inhibition in BT-549 and MDA-MB-468. We found a similar result in MDA-MB-231 basal cell line treated with *ZNF703*-ASO9.

Other studies using siRNA showed that knockdown of *ZNF703* expression inhibited papillary thyroid carcinoma, medullary thyroid carcinoma, and colorectal cancer cell proliferation and migration [43,48,49] comforting *ZNF703* as an oncogene and a potential therapeutic target for other advanced cancers.

### 4.3. ZNF703 mRNA-Targeting May Be Used in Combination with Chemo- and Hormone-Therapy of Advanced Breast Cancers

After BC treatment, surviving cancer cells cause metastasis, which remains the main cause of cancer-related mortality.

Clinical trials have shown that cisplatin treatment (alone or in combination with other anticancer drugs) is efficient in inducing BC cell death and decreasing tumor volume [75,76]. These findings have renewed interest in cisplatin and other chemotherapies sharing ancestry with cisplatin as a therapy for BC. However, several mechanisms may explain BC resistance against cisplatin [77].

Here, we report that the anti-cancer ability of *ZNF703*-ASO9 was improved when it is combined with cisplatin in the luminal B model MCF7-*ZNF703*/GFP.

Because luminal B BC patients after several lines of treatment without success present a poor chance of survival, the smallest improvement may be appreciated. Strategies such as ASO9/cisplatin combination could be considered as a targeted therapy and may thus constitute a real hope for luminal advanced BC patients.

For high-risk, ER+, HER2− BC, standard adjuvant anthracycline-taxane regimens are appropriate when neoadjuvant chemotherapy is chosen. However, optimal hormone therapy is the standard of care in the adjuvant setting, whether or not a pathological complete response is obtained [78]. Tamoxifen is commonly used in the treatment of luminal BC. However, half of patients treated with tamoxifen are insensitive. Luminal BC cell lines overexpressing *ZNF703* were reported resistant to tamoxifen through activation of AKT/MTOR signaling [79]. Overexpression of *ZNF703* in MCF7 luminal BC cells induced activation of the AKT/MTOR signaling pathway, downregulation of ERα, and reduction of the tamoxifen’s antitumor effect. While low-dose tamoxifen stimulated the growth of cells overexpressing *ZNF703*, treatments of tamoxifen-treated MDA-MB-134 and HCC1500 luminal B BC cell lines with (i) *ZNF703*-siRNA alone significantly reduced survival rates whereas the combination *ZNF703*-siRNA with the MTOR inhibitor rapamycin enhanced the antitumor effect of tamoxifen [79]. This information suggests that the next step to evaluate the ability of *ZNF703*-ASO9 treatment to restore hormone sensitivity in luminal BC could be to use tamoxifen combined with an MTOR inhibitor. Recently, an ASO-targeting circPVT1 was reported to inhibit ER+ cell and tumor growth, re-sensitizing tamoxifen-resistant ER+ BC cells to tamoxifen treatment [80].

Cyclin-dependent kinase 4 and 6 (CDK4/6) inhibitors (e.g., palbociclib, ribociclib) have shown significant efficacy in ER+ BC [24,25]. Therapy designs combining *ZNF703*-ASO9 and these inhibitors should also be considered.

## 5. Conclusions

*ZNF703* overexpression facilitates tumorigenesis, metastatic invasion, and predicts poor prognosis in luminal B BCs but also in other advanced cancers. *ZNF703* inhibition may be a potential targeted treatment for such advanced cancers. Our work shows that ASO technology is a way to efficiently inhibit *ZNF703* in luminal B BC cells. This inhibition decreased cell proliferation, provoked apoptosis and the combination with cisplatin improved the anti-cancer ability of *ZNF703*-ASO9 in the luminal B model MCF7-*ZNF703*/GFP. ASO-based inhibition of *ZNF703* may be useful in the treatment of luminal B BCs as well as other advanced cancers either as a monotherapy or in combination with other therapies.

## Figures and Tables

**Figure 1 pharmaceutics-15-01930-f001:**
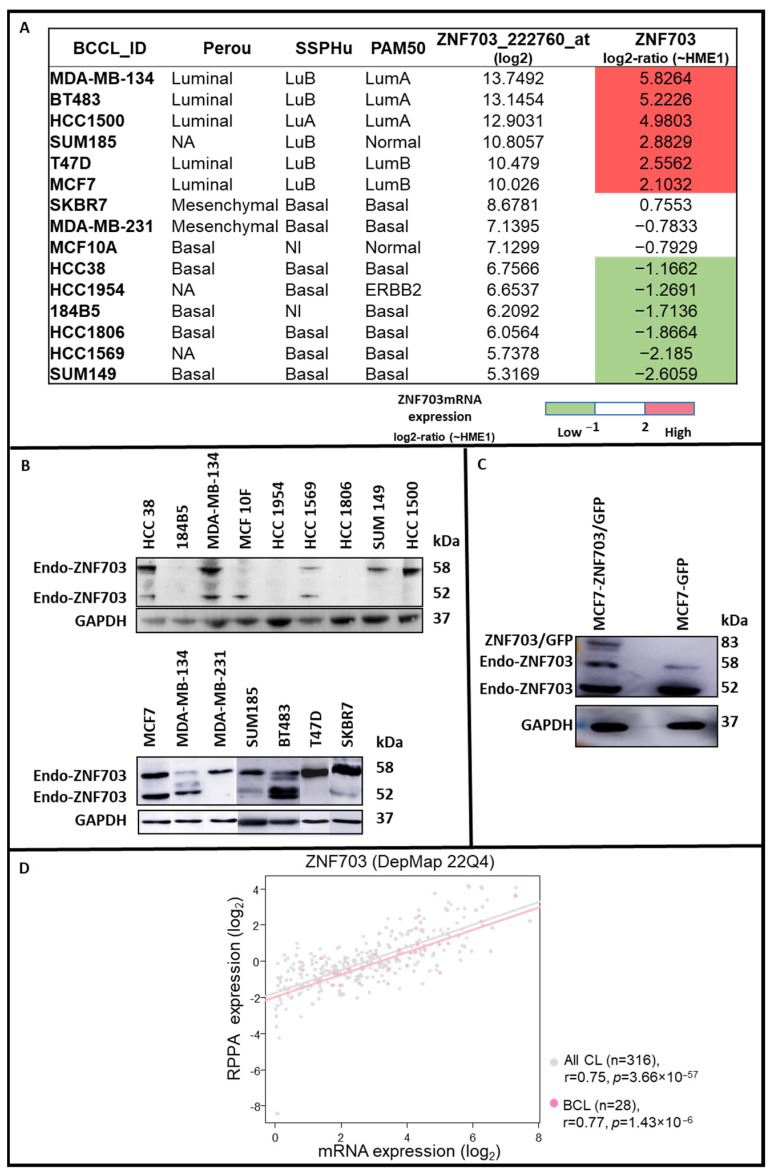
*ZNF703* mRNA and *ZNF703* protein expression in breast cancer cell lines. 15 BC cell lines were chosen for their *ZNF703* mRNA expression centered on that observed in the HME-1 cell line and defined as follows: (i) top 6 of upregulated (High); (ii) 3 medium (Med) and (iii) top 6 of down-regulated (Low). For each of them, a molecular subtype was established with Perou, SSP, and PAM50 classifications [62,65,66] (**A**). The *ZNF703* protein expression was assessed on lysates of the 15 breast tumor cell lines and flow cytometry sorted MCF7-*ZNF703*/GFP and MCF7-GFP cells (used as controls) by western blotting (WB) ((**B**,**C**), respectively). The two endogenous *ZNF703* protein isoforms (58 and 52kDa) (Endo-*ZNF703*) were variably observed in cell lines. Only the full-length 58kDa *ZNF703* isoform was approved by the CCDS database (https://www.ncbi.nlm.nih.gov/CCDS/CcdsBrowse.cgi?REQUEST=CCDS&DATA=CCDS6094 (accessed on 15 May 2022); Protein identity/Ref Sequence: NP_079345.1) and considered as previously reported [41,42,43]. An additional band corresponding to *ZNF703*/GFP fusion protein (83 kDa) was only observed in MCF7-*ZNF703*/GFP. High *ZNF703* expression was observed in BT483, HCC1500, MCF7, SKBR7, and T47D. The *ZNF703* mRNA and *ZNF703* protein expression data collected from 316 cancer cell lines, including 28 breast cancer cell lines, [67] were analyzed and compared. A significant positive correlation between the mRNA and protein expression levels of *ZNF703* was established in all cell lines (Pearson r = 0.75, *p* = 3.66 × 10^−57^) and breast cell lines (Pearson r = 0.77, *p* = 1.43 × 10^−6^) (**D**).

**Figure 2 pharmaceutics-15-01930-f002:**
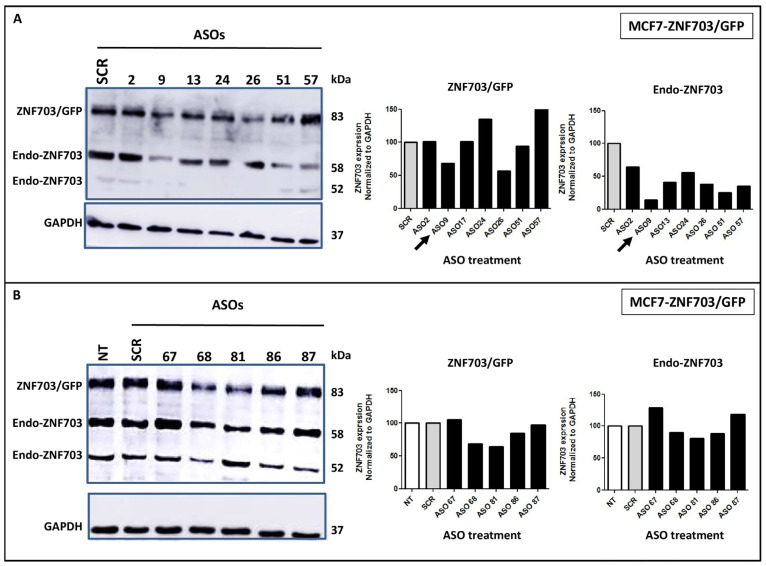
***ZNF703* ASOs screening for inhibitory activity on *ZNF703* mRNA expression.** (**A**,**B**). *ZNF703* expression is assessed by WB on protein lysates extracted from engineered MCF7-*ZNF703*/GFP BC cells treated with 100 nM concentration of various designed ASOs. WB images showing *ZNF703* band signals intensities were respectively quantified for *ZNF703*/GFP (83 kDa) and endo-*ZNF703* (58 kDa) by Image J, normalized to GAPDH, and represented as bars. ASO9 was identified to show higher downregulation efficiency and effectively represses both endogenous and exogenous *ZNF703* (**B**).

**Figure 3 pharmaceutics-15-01930-f003:**
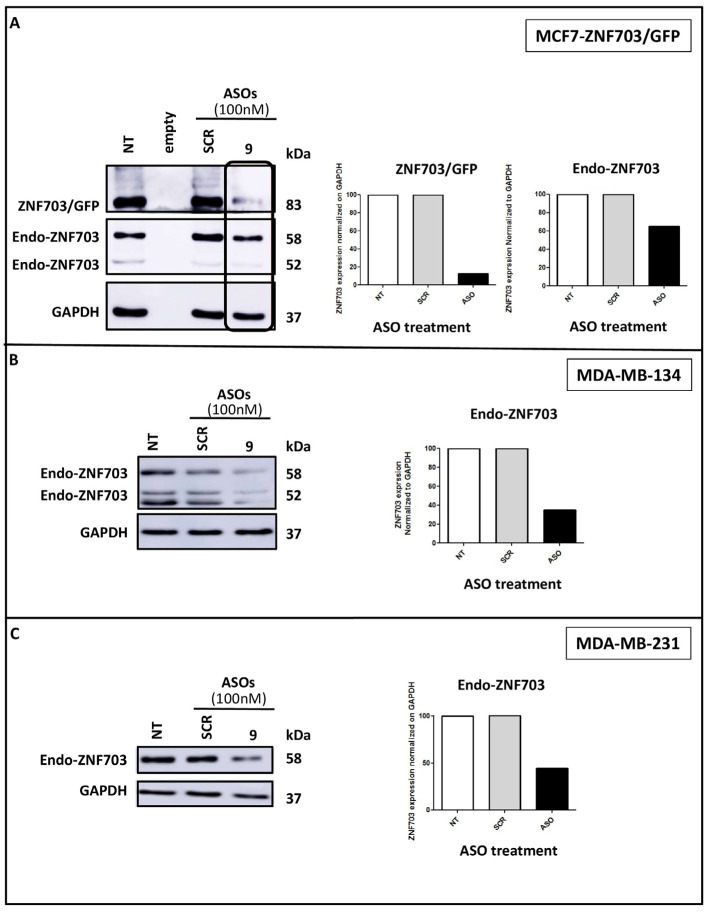
**ASO9 downregulates *ZNF703* protein in BC cell lines.** Western blot analysis showed *ZNF703* inhibition with ASO9 in engineered MCF7-*ZNF703*/GFP and MCF7-GFP, MDA-MB-134, and MDA-MB-231 BC cell lines ((**A**,**B**,**C**), respectively). WB images showing *ZNF703* band signal intensities were respectively quantified for *ZNF703*/GFP (83 kDa) and endo-*ZNF703* (58 kDa) by Image J, normalized to GAPDH, and represented as bars. While ASO9 downregulated both endogenous and exogenous *ZNF703* expression in MCF7-*ZNF703*/GFP (**A**), a lower expression inhibition was observed in MDA-MB-134 and MDA-MB-231 cell lines ((**B**,**C**), respectively).

**Figure 4 pharmaceutics-15-01930-f004:**
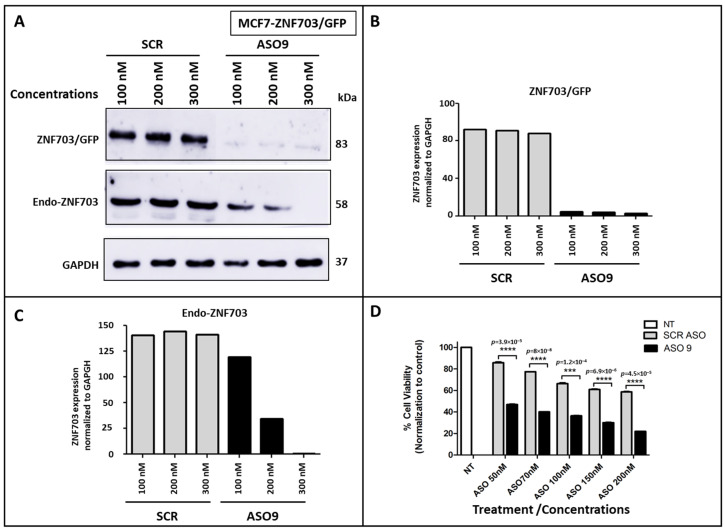
ASO9 reduces *ZNF703* expression and viability in MCF7-*ZNF703*/GFP BC cells. MCF7- *ZNF703*/GFP cells were treated with ASO9 or control-ASO (scrambled, SCR) at various concentrations. *ZNF703* expression inhibition was analyzed by WB (**A**). WB images showing *ZNF703* band signals intensities were respectively quantified for *ZNF703*/GFP (83kDa) and endo-*ZNF703* (58 kDa) by Image J, normalized to GAPDH, and represented as bars. The *ZNF703*/GFP (83kDa band) expression was quickly inhibited from 100nM dose of ASO9 (**A**,**B**) while the endogenous *ZNF703* band at 58 kDa, was inhibited by ASO9 following a dose-dependent manner (**A**,**C**). The cellular toxicity effect of *ZNF703* knockdown induced by ASO9 in MCF7-*ZNF703*/GFP cells was measured with MTT assay. ASO9 decreased the number of viable cells in a dose-dependent manner, with the cell viability rate of control-ASO and ASO9 reduced to 62.247 ± 5.508 and 21.913 ± 0.090 at 200 nM, respectively (Student’s *t*-test, *** *p* < 0.001, and **** *p* < 0.0001) (**D**).

**Figure 5 pharmaceutics-15-01930-f005:**
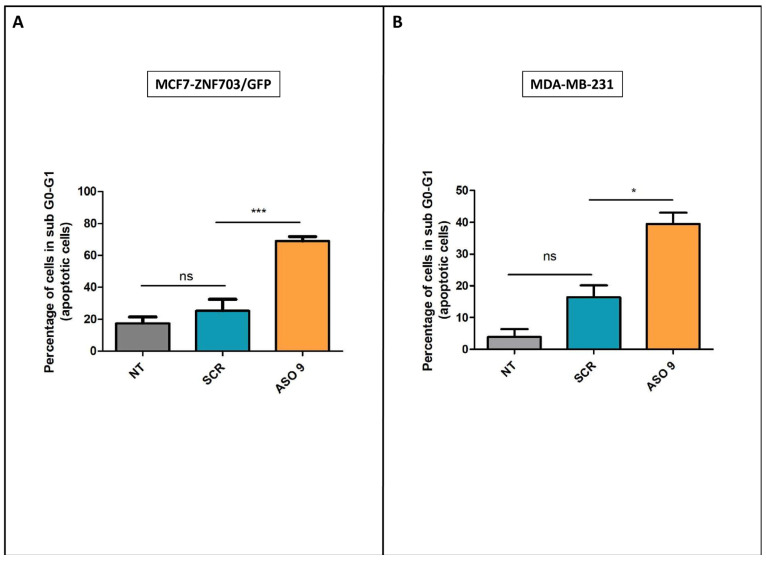
**ASO9 induces apoptosis in MCF7-*ZNF703*/GFP and MDA-MB-231 BC cell lines.** A cell apoptosis test by annexin V binding was done in MCF7-*ZNF703*/GFP and MDA-MB-231 BC cell lines ((**A**,**B**), respectively). Flow cytometry analysis of annexin V expression showed that treatment with ZNF-ASO9 significantly increased apoptosis of MCF7-*ZNF703*/GFP cells (*** *p* ≤ 0.001) and MDA-MB-231 cells (* *p* ≤ 0.05) compared to control-ASO. MCF7-*ZNF703*/GFP and MDA-MB-231 cells not treated (NT) and treated with control-ASO showed a non-significant difference in the percentage of annexin V positive cells, suggesting low toxicity of control-ASO.

**Figure 6 pharmaceutics-15-01930-f006:**
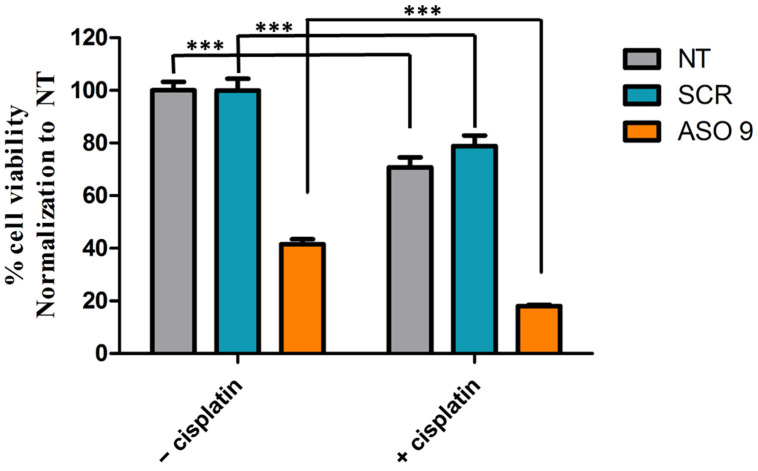
The combination with cisplatin improves the anti-cancer ability of *ZNF703*-ASO9 in MCF 7-*ZNF703*/GFP BC cell line. To test the efficacy of cisplatin combination with ASO9 treatment, we inhibited *ZNF703* with ASO9 (100 nM) and treated MCF7-*ZNF703*/GFP BC cell line with cisplatin (100 nM) 48hrs post ASO9 transfection. The experiment was done in triplicate and cell viability was evaluated by the MTT test. The results showed that: (i) cisplatin alone or in combination with SCR reduced cell viability more efficiently than the absence of treatment (NT) or with SCR alone (Student’s *t*-test, *** *p* ≤ 0.001); (ii) *ZNF703* inhibition by ASO9 in combination with cisplatin reduced cell viability more efficiently than treatment with ASO9 alone or cisplatin alone (Student’s *t*-test, *** *p* ≤ 0.001).

## Data Availability

All data are contained in the article.

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
