# Peer review of "ZNF703 mRNA-Targeting Antisense Oligonucleotide Blocks Cell Proliferation and Induces Apoptosis in Breast Cancer Cell Lines"

_pharmaceutics, 2023, doi:10.3390/pharmaceutics15071930_

Round 1
Reviewer 1 Report
This manuscript analyzes the expression of ZNF703 in different breast cancer cell lines and designs mRNA-targeting antisense oligonucleotides for downregulating endogenous and exogenous ZNF703 expression in breast cancer cell lines. ZNF703 ASO can also effectively inhibit the survival rate of tumor cells. However, the author did not choose a suitable cell line for research. The cell line selected by the author is not representative. The author should choose a cell line that can clearly express the two isoforms for the experiment showing the down-regulation effect of ZNF703. And authors need to select drug-resistant cell lines to study whether ZNF703 downregulation can increase the sensitivity to cisplatin resistance.
1. As mentioned by the author, there are two isoforms of ZNF703, and the results of Western also show the bands of the two isoforms. Which isoforms work during treatment? Which isoform did the mRNA target, and why choose this isoform?
2. MDA- MB-231 only shows 58kDa ZNF703? MDA-MB-134 expresses more 52kDa ZNF703 than 58kDa, while the result shows most cell lines express more 58kDa ZNF703. Are they representative? Why did the authors choose these cell lines for further experiments?
3. In Figure 4D, please add the p-value between ASO groups in different concentrations.
4. In Figure 7, MCF 7-ZNF703/GFP BC cell line lines are not drug-resistant cells, and cannot indicate whether this cell line is resistant to cisplatin, so ASO cannot indicate enhanced cisplatin chemosensitivity at all. After adding cisplatin, the NT and SCR groups reduced the cell survival rate by 20%-30%, while the ASO group reduced the cell survival rate by 20%. Your results can only show that ASO can reduce the cell survival rate very well, irrelevant with cisplatin drug resistance. Authors need to choose drug-resistant cell lines for research, such as MCF7/ADR and MDA-MB-231/DDP.
5. Figure 5 can be moved to supplementary.
Reviewer 2 Report
Authors developed an antisense oligonucleotide (ASO) against ZNF703 mRNA encoding a protein whose overexpression may contribute to drug resistance against endocrine drugs. They also shown that selected ASO downregulates efficiently ZNF703 protein expression. I advise publication of this work with minor corrections.
Please comment on possible reasons of various effect on cell viability and ZNF703 protein level of different designed ASOs.
In Material and methods. Design and Synthesis of Antisense Oligonucleotides, please specify the length of synthesized ASOs.
Round 2
Reviewer 1 Report
As the authors are not sure whether MCF7-ZNF703/GFP is drug-resistant (our results show that MCF7-ZNF703/GFP is sensitive to cisplatin), and the authors do not want to use any drug-resistant cell lines for the study, so the article does not prove that ASO9 enhances cisplatin sensitivity. The author's experimental results can only show that ASO9 has anti-cancer effects, and it improves when combined with cisplatin. As for the author's mention of drug resistance in clinical patients, the author did not conduct any research or explanation. It is recommended that the author delete the words about drug resistance and only mention the anticancer ability of ASO9
